# Understanding Where We Are Well: Neighborhood-Level Social and Environmental Correlates of Well-Being in the Stanford Well for Life Study

**DOI:** 10.3390/ijerph16101786

**Published:** 2019-05-20

**Authors:** Benjamin W. Chrisinger, Julia A. Gustafson, Abby C. King, Sandra J. Winter

**Affiliations:** 1Department of Social Policy and Intervention, University of Oxford, Oxford OX1 2ER, UK; 2Stanford Prevention Research Center, Department of Medicine, Stanford University School of Medicine; Stanford, CA 94305, USA; julia.gustafson@stanford.edu (J.A.G.); king@stanford.edu (A.C.K.); sjwinter@stanford.edu (S.J.W.); 3Department of Health Research and Policy, Stanford University School of Medicine, Stanford, CA 94305, USA

**Keywords:** well-being, wellness, geography, neighborhoods, spatial analysis, social determinants of health

## Abstract

Individual well-being is a complex concept that varies among and between individuals and is impacted by individual, interpersonal, community, organizational, policy and environmental factors. This research explored associations between select environmental characteristics measured at the ZIP code level and individual well-being. Participants (*n* = 3288, mean age = 41.4 years, 71.0% female, 57.9% white) were drawn from a registry of individuals who completed the Stanford WELL for Life Scale (SWLS), a 76-question online survey that asks about 10 domains of well-being: social connectedness, lifestyle and daily practices, physical health, stress and resilience, emotional and mental health, purpose and meaning, sense of self, financial security and satisfaction, spirituality and religiosity, and exploration and creativity. Based on a nationally-representative 2018 study of associations between an independent well-being measure and county-level characteristics, we selected twelve identical or analogous neighborhood (ZIP-code level) indicators to test against the SWLS measure and its ten constituent domains. Data were collected from secondary sources to describe socio-economic (median household income, percent unemployment, percent child poverty), demographic (race/ethnicity), and physical environment (commute by bicycle and public transit), and healthcare (number of healthcare facilities, percent mammogram screenings, percent preventable hospital stays). All continuous neighborhood factors were re-classified into quantile groups. Linear mixed models were fit to assess relationships between each neighborhood measure and each of the ten domains of well-being, as well as the overall SWLS well-being measure, and were adjusted for spatial autocorrelation and individual-level covariates. In models exploring associations between the overall SWLS score and neighborhood characteristics, six of the twelve neighborhood factors exhibited significant differences between quantile groups (*p* < 0.05). All of the ten SWLS domains had at least one instance of significant (*p* < 0.05) variation across quantile groups for a neighborhood factor; stress and resilience, emotional and mental health, and financial security had the greatest number of significant associations (6/12 factors), followed by physical health (5/12 factors) and social connectedness (4/12 factors). All but one of the neighborhood factors (number of Federally Qualified Health Centers) showed at least one significant association with a well-being domain. Among the neighborhood factors with the most associations with well-being domains were rate of preventable hospital stays (7/10 domains), percent holding bachelor’s degrees (6/10 domains), and median income and percent with less than high school completion (5/10 domains). These observational insights suggest that neighborhood factors are associated with individuals’ overall self-rated well-being, though variation exists among its constituent domains. Further research that employs such multi-dimensional measures of well-being is needed to determine targets for intervention at the neighborhood level that may improve well-being at both the individual and, ultimately, neighborhood levels.

## 1. Introduction

Community characteristics matter when it comes to individuals’ well-being, and that relationship is better understood now than ever before [1]. Several studies have demonstrated this [2,3] and knowing how these community characteristics can influence holistic well-being brings a new opportunity for communities to intervene with policies and practices that broadly enhance well-being. Despite the important implications of such information, debates continue over the most appropriate measures of well-being.

Determining the appropriate spatial scales at which human well-being may be influenced by the built and social environment is another challenge facing researchers. It is important to consider, for example, how measures of access or opportunity may be appropriate for certain neighborhood characteristics (e.g., number of alcohol or tobacco retailers in a given geography, or availability of neighborhood parks), while measures of exposure are needed for others (e.g., amount of advertising for alcohol and tobacco products in a given geography, or distance to a green open space) [4]. Additional challenges arise when considering the potential sorting of individuals into neighborhoods that might suit their preexisting behaviors or traits [5]. In the case of well-being, the exact pathways linking person and place are not well understood, perhaps, in part due to our conceptualization and measurement of well-being.

While it is suggested that the measurement of well-being is multi-dimensional [6], previous studies examining relations between well-being and neighborhood contexts have primarily used selections of well-being survey instruments, such as the Gallup-Sharecare Well-being Index (WBI), the 36-Item Short Form Survey (SF-36), or the General Social Survey (GSS) [2,7,8,9]. While other studies that have utilized the entirety of the survey instruments, they mostly focus on well-being as the overall global score of the instruments, overlooking the potential influence of the environment on the individual domains that comprise the overall scores.

While global measures of well-being encourage a conceptualization of human health that is broader than physical ability or illness-related dimensions, they do not offer much insight into exactly how well-being could be related to the neighborhood context. Psychological research has illustrated how multi-component measurement tools are needed to account for variation across individuals’ experiences of well-being and its three broad dimensions: [10] 1) hedonic, which focuses on individuals’ happiness and enjoyment; 2) evaluative, which focuses on individuals’ satisfaction with their life; and 3) eudaimonic, which focuses on individuals’ sense of purpose and meaning [11]. Yet, what is missing from such conceptualizations is how different elements of well-being may be linked to an individual’s neighborhood context.

The Stanford WELL for Life Initiative has created a more comprehensive measure of well-being, the Stanford WELL for Life Scale (SWLS) [10], that measures ten constituent domains (described in Section 2.1). These domains collectively address the above three dimensions of well-being and its pluralistic nature. The SWLS provides participants with ten separate domain scores that, when combined, provide an overall well-being score. Thus, the SWLS can provide a more nuanced look at how the many dimensions of well-being are associated with surrounding community characteristics at the neighborhood (in this case, ZIP-code) level.

In this study, we first compare the global SWLS measure, collected in a sample of U.S. adults who registered online for the WELL for Life Initiative, to a recent nationally-representative study (*n* = 338,846) by Roy and colleagues [8] that examined well-being as measured by the Gallup-Sharecare Well Being Index individual well-being score, or iWBS, and 77 county-level physical, social, and demographic factors. That study found independent and significant associations between well-being and twelve county-level indicators, some of which are used in generating scores for the well-known Robert Wood Johnson Foundation County Health Rankings [12]. We draw upon the same data sources to carefully mirror the indicators used by Roy et al., but at a more fine-grained geographic unit, the ZIP code. Subsequently, we test for significant associations between the SWLS domain measures and these 12 indicators to further investigate relationships between place and well-being. In both parts of this investigation, we use ZIP-code level analogues to the county-level factors described in Roy et al.

## 2. Materials and Methods

### 2.1. Outcome Variables: Well-Being and Its Domains

The Stanford WELL for Life registry was started in May 2016 to accelerate the science of well-being and to improve and sustain health and well-being. Data collection is ongoing and for the purposes of this analysis, data were drawn on September 15, 2018, and the sample includes 3611 adult participants. To be eligible to participate in the registry, individuals had to be aged 18 years and older, residing in the United States, and able to complete the online survey in English, Spanish or simplified or traditional Chinese. Participants were recruited via existing research registries and email list serves within Stanford; on-line through social media, e-mail blasts and webpages; through existing community partnerships that assisted with targeted recruitment strategies aimed at populations of interest, for example, working with Asian community-based collaboratives to recruit Asian participants; and at community events such as health fairs.

The Stanford WELL for Life Scale (SWLS) was used to measure individual level well-being. This scale asks respondents to rate their well-being for the past two to four-week time period. The SWLS was developed from qualitative data gathered using a grounded narrative approach in which individuals described times of high and low well-being, with no priming regarding the definition of well-being [10]. Ten domains of well-being were identified, as follows: social connectedness, lifestyle and daily practices (diet, physical activity, sleep, tobacco and alcohol use), experience of emotions, stress and resilience, physical health, purpose and meaning, sense of self, financial security and satisfaction, exploration and creativity, and spirituality and religiosity. To measure the ten domains of well-being identified during the qualitative data gathering and analysis, the quantitative SWLS was developed using previously validated questions where relevant and available, or questions created de novo by the research group to address gaps. Each of the ten domains is scored from 0–10, and an unweighted overall well-being score is calculated by summing each of the ten domain scores. The lifestyle and daily practices domain contains five sub-domains (diet, physical activity, sleep, tobacco, and alcohol use) which each contribute up to 2 points to the lifestyle and daily practices domain. For the purposes of this analysis, all well-being outcomes were treated as continuous variables.

All data preparation, transformation, and analyses were performed using R version 3.5.1 [13].

### 2.2. Sociodemographic Covariates

Additional data gathered from participants included socio-demographic information (age, gender, education, household income, employment status, and marital status), household information (number of individuals living in the household, how many times individuals have moved in the past year, number of children, and where they live), and health history and status (access to healthcare, diagnosis of a variety of chronic, and acute conditions).

### 2.3. Exposure Variables: Neighborhood-Level Social and Environmental Features

To be included in the Stanford Well for Life study participants had to provide a valid 5-digit ZIP-code, excluding post-office boxes. This geographic identifier was then used to assign neighborhood-level data to each participant. We selected 12 ZIP-level variables (see Table 1) that were as similar as possible to the physical environment, social/economic, and demographic factors found to be significant in a recent empirical study of 77 potential county-level correlates of well-being [8]. Five-year estimates of the 2013–2017 American Communities Survey were used for ZIP code-level demographic characteristics (percent Black or African American), educational attainment (three values: 9–12 grade without a diploma, high school diploma or equivalent, and bachelor’s degree), median household income, divorce rate, child poverty rate, unemployment rate, and commuter characteristics (percent commuting by public transit, percent commuting by bicycle). Similar to the County Health Rankings used by Roy et al. [8,12], we drew health care-related data from the 2015 Dartmouth Atlas of Health Care (DAHC) [14]. These included the percent of women receiving mammograms and percentage of preventable hospital stays, which were summarized at the HSA (hospital service area) level and re-aggregated to ZIP codes based on an Atlas-provided crosswalk. ZIP codes having at least one Federally Qualified Health Center (FQHC) were identified from a US Health Resource and Services Administration (HRSA) database of health center service delivery sites [15]. Table 1 provides a summary of neighborhood-level data used in this study.

### 2.4. Data Transformations

As an exploratory, hypothesis-generating exercise, this project sought to understand how relative (rather than absolute) differences in neighborhood factors correlated with well-being outcomes. To facilitate this, continuous neighborhood factor variables were transformed into roughly equal quantile groups, with the lowest quantile serving as the reference level in analyses, again following the method used in the comparison study by Roy and colleagues [8]. Given the geographic distribution of participants across ZIP-codes, roughly equal quantile groups were not feasible for two variables (% mammography and # FQHCs); in these scenarios, alternative groupings were used. A summary of all quantile group characteristics is provided in Appendix A
Table A1.

Tukey’s Ladder of Powers transformations [16] were applied to all well-being outcome variables using the “tranformTukey” function of the “rcompanion” R package [17,18], and the attendant lambda values were used to back-transform coefficient estimates and standard errors for reporting.

### 2.5. Statistical Models

Linear mixed models fit by maximum likelihood were generated to assess the relationship between well-being outcomes (the overall SWLS measure and ten domains) and the previously-mentioned 12 neighborhood and individual-level covariates [19]. To account for spatial aggregation of neighborhood data, a ZIP-level grouping variable was incorporated as a random effect. For each model, participants with missing data were omitted and assumed to be missing at random. Spatial autocorrelation of the primary outcome variable was assessed at the level of participants (individual well-being scores) and using Morans’ *I* test statistic for spatial autocorrelation [20,21], which indicated significant geographic clustering of similar outcome values. To account for this spatial autocorrelation of the outcome variable, a Gaussian correlation structure was specified using latitude and longitude coordinates of ZIP code centroids. Spatial autocorrelation was then assessed with Moran’s I test for residuals from the full linear mixed model for the main well-being outcome. All linear mixed modeling was performed using the “lme” function from the “nlme” R package [22,23,24,25], and Moran’s I tests were performed using the “moran.test” function from the “spdep” R package [26]. These spatial statistics and their attendant p-values are reported in Tables 5 and 6.

Two models were constructed for the overall SWLS well-being score and its ten constituent domains: a “partial” model only including neighborhood factors and a “full” model with neighborhood and individual-level covariates. Backward and forward stepwise model selection by Akaike Information Criterion (AIC) was performed using maximum likelihood parameter estimation to identify covariates for inclusion in final models, using the “stepAIC” function from the “MASS” R package [27]. Prior to this procedure, outliers (defined as observations having a standardized residual distance greater than 2.5 from 0) were identified based on the initial partial and full model fits and removed [28]. With the covariates identified in stepwise model selection, partial and full models were refit using restricted maximum likelihood parameter estimation, to partially account for the unbalanced nature of our higher-level groups (i.e., ZIPs). To assess the amount of variance explained by the ZIP code groupings, intra-class correlation coefficients were calculated, and marginal and conditional r-squared values were calculated to assess variance with and without the ZIP-level random effect [29,30,31]. Finally, to identify significant differences in coefficient estimates between quantile groups of each neighborhood factor, type II sum of squares test for ANOVA effects were performed using the “car” R package’s built-in “Anova” function [32].

## 3. Results

### 3.1. Participant and Neighborhood Characteristics

The initial SWLS dataset included 3611 unique participants. Participants who did not provide a valid ZIP code (*n* = 41), provided a post-office box address (*n* = 110), or did not complete enough of the SWLS survey as to generate an overall well-being score (*n* = 232) were excluded from this analysis. In total, 3288 participants were included in this study. The participant population was comprised of 71.0% women and was, on average, 41.4 years old (SD = 17.6, min = 18, max = 94). The majority of participants identified as white (57.9%), currently working (64.4%), and earning relatively high incomes (52.7% earning over $100k per year). Many participants also reported being married (42.1%) and not having moved in the last five years (42.8%). Table 2 provides a summary of participant characteristics and average well-being scores by demographic category.

Participants were from a total of 485 unique ZIP-codes representing 152 separate counties across 36 states. California had the vast majority of participants (93.2%), and the average ZIP-code had 7.1 participants (SD = 31.8, min = 1, max = 584). Table 3 describes the average neighborhood characteristics according to the 12 neighborhood factors used in our statistical models, as well as the valid number of observations. Figure 1 illustrates how average domain scores vary by quantile group of each neighborhood factor.

### 3.2. Overall Well-Being and Neighborhood Factors

The overall SWLS measure of well-being was significantly (*p* < 0.05) associated with six of the 12 neighborhood factors identified by Roy and colleagues using the iWBS. Among these factors, consistently higher well-being outcomes across neighborhood factor quantiles (quantile groups 2–5 versus Q1) were identified for percent with a bachelor’s degree (*p* = 0.045) and median income (*p* = 0.001). Conversely, higher quantiles according to percent with partial high school education (*p* = 0.019), mammography rate (*p* = 0.035), and rate of preventable hospital stays (*p* = 0.001) had consistently lower SWLS estimates compared to the lowest quantile group. Table 4 provides coefficient estimates for each neighborhood factor by quantile group based on a model without participant socio-demographic characteristics as covariates. In a model adjusted for individual-level covariates (see Table 6), the association between percent commuting by public transit and well-being is no longer significant, though it still contributes to a model that comparatively better fits the data, compared to a model without it (as measured by AIC values). The full model output, including coefficients and p-values for the individual-level covariates, is available in Appendix B
Table A2.

### 3.3. Domains of Well-Being and Neighborhood Factors

All of the ten well-being domains had at least one instance of significant variation across a quantile groups for a neighborhood factor (see Table 5 for outcomes without individual-level controls, and Table 6 for fully-adjusted outcomes). Of these, the stress and resilience, emotional and mental health, and financial issues domains had the greatest number of significant associations (six of 12 neighborhood factors) in models without individual-level adjustments, followed by physical health (five of 12 factors), and social connectedness (four of 12 factors). The purpose and meaning domain had only one significant association with neighborhood factors in unadjusted models. The number of FQHCs had no significant associations with any well-being domain, though it was included in two unadjusted domain models: creativity and exploration and stress and resilience. Among the neighborhood factors with the most associations with well-being domains was rate of preventable hospital stays (seven of ten domains), percent holding bachelor’s degrees (six of ten domains), and median income and percent with less than high school completion (five of ten domains).

When controlling for individual-level covariates, some, but not all, of the associations between neighborhood factors and well-being domains lose their statistical significance (see Table 6). All but one neighborhood factor (number of FQHCs) were included and statistically significant in fully-adjusted final models for at least one well-being domain, with percent of partial high school education maintaining significance across the most domains (five of 10 domains). All well-being domains, except for purpose and meaning, maintained at least one significant association with a neighborhood factor after individual-level covariates were included.

## 4. Discussion

While using an independently-developed and unique measure of well-being, we found similar associations between five of the twelve county-level factors identified by a much larger, nationally-representative survey of well-being (iWBS) [8]. Similar to iWBS, our SWLS measure of well-being was significantly associated with socio-economic indicators such as educational attainment and median household income, clinical characteristics; as measured by the percent of hospital stays classified as “preventable” and mammography rate, and physical environment characteristics, as defined by the commuting patterns of residents. The overall SWLS well-being measure was not significantly associated with six of the significant factors in the Roy et al. study—divorce rate, child poverty rate, mammography rate, number of FQHCs and commuting by bicycle—though of these, only the number of FQHCs did not exhibit any significant associations with the constituent SWLS domains.

Even when adjusting for major, well-studied individual-level characteristics, like gender, income, and employment and marital status (all of which exhibited significant effects on SWLS and its domains), statistically significant variation between quantile groups of the neighborhood factors remained. While the insights given by Roy and colleagues in their nationally-representative study provide an important focus on key neighborhood-level associations with well-being, this paper deepens that understanding by affirming many of their findings with an independently-derived well-being measure. Additionally, by further identifying associations with specific domains of well-being, we provide a basis for additional inquiries into the nature of these relationships between place, health, and well-being.

The results can set the stage for the development of interventions in the physical environment arena as a means for enhancing well-being, including health behaviors that can impact it [8]. While “top-down” policy-level approaches to environmental change have often been targeted, an emerging area of “bottom-up” community engagement activities around changing local physical environments to support healthy living have shown increasing promise [33]. Community-engaged research models such as the Our Voice global citizen science research initiative [34] may provide fruitful methods for eliciting and identifying possible mechanisms, supports, and barriers to achieving the types of well-being outcomes targeted in this study [35].

Strengths of this study include its multi-dimensional measure of well-being, as well as a more fine-grained treatment of place (i.e., at the ZIP level). By exploring relations of both a global well-being score and its component parts, we are able to develop hypotheses about which aspects of well-being are associated with different aspects of the built and social environment. For example, if it is possible that some domains of well-being are relatively insulated to the effects of neighborhood environment, while others are quite sensitive, or that individuals self-select into certain neighborhoods based on amenities that support their well-being (e.g., that can directly impact the SWLS domain of lifestyle and daily practices). By conducting our investigation at a smaller geographic level, we may better identify likely exposures to the kinds of environments that may be too aggregated at the level of counties and states. For example, participants residing in Santa Clara County, California, were spread across 55 separate ZIP codes, which ranged widely in neighborhood characteristics. Finally, by assessing differences across quantile groupings of neighborhood factors (as in Roy et al. [8]), rather than seeking to draw more precise “dose-response” inferences, we allowed for the possibility that certain effects of neighborhood characteristics on well-being are non-linear (e.g., both extremely low and extremely high values exhibited a positive relationship with well-being, while mid-range values had a negative relationship).

This study also has several limitations. Several data assumptions were made that are important limitations. First, while SWLS domain scores are treated as continuous for the purpose of this exploratory study, some domains were generated from single questions, making them ordinal in nature, which may be more fully investigated in future studies. Second, we assume data to be missing at random, which may be a source of bias should significant patterns of omission exist. Future studies might incorporate weighting schemes based on participant non-response, which is not currently a feature of the SWLS dataset. The study’s cross-sectional design also precludes our ability to draw causal inferences based on the significant correlations we have described. Future research utilizing longitudinal cohort studies, including the Stanford WELL for Life registry, will enable additional investigations of these relations as both individual-level variables and neighborhood contexts change over time. Longitudinal studies may also track individuals who move from one neighborhood context to another, offering yet another opportunity to account for exposure. Another limitation of the study is the difficulty of dealing with self-selection bias. Common to neighborhood research, we are not able to rule out the possibility that individuals “sort” themselves into neighborhoods that reinforce pre-existing individual characteristics or traits, rather than the scenario suggested here, whereby neighborhood contextual factors may influence different aspects of an individual’s self-rated well-being. One final limitation is related to individual exposure to neighborhood contexts; for example, while each participant is assigned a summary score for a neighborhood characteristic, an individual’s unique exposure to, or experience with that factor, may be quite variable. Focusing our analysis at the sub-county level helps address this question to some degree, although it does not entirely control for the potentially confounding effects of this dynamic.

## 5. Conclusions

This study sought to test recently-identified contextual factors associated with well-being by applying a novel measure of well-being and finer-grained, sub-county level of aggregation. Our findings affirm many of the previously-identified associations between well-being and place, including those related to demographic and socioeconomic characteristics. Certain domains of well-being such as lifestyle and daily practices and emotional and mental health, appear to be more sensitive to neighborhood factors than others, like purpose and meaning or exploration and creativity. More exploration is needed into the nature of these relationships, including the possible mechanisms underlying the relations between constituent domains of well-being and place.

## Figures and Tables

**Figure 1 ijerph-16-01786-f001:**
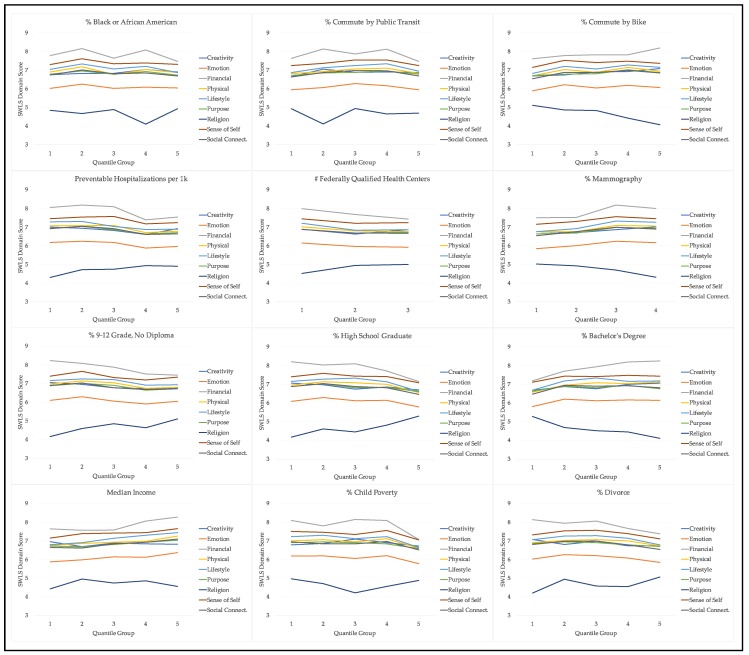
Average SWLS domain scores by neighborhood factor quantile group.

**Table 1 ijerph-16-01786-t001:** Summary of 12 Neighborhood-Level Factors and Data Sources.

Neighborhood Factor	Source	Level
Race: % Black or African/American	ACS, 2013–2017	ZIP
Education: % 9–12 grade, no degree	ACS, 2013–2017	ZIP
Education: % high school diploma/GED	ACS, 2013–2017	ZIP
Education: % bachelor’s degree	ACS, 2013–2017	ZIP
% Divorce	ACS, 2013–2017	ZIP
Median household income	ACS, 2013–2017	ZIP
% Children in poverty (% households with children below poverty line)	ACS, 2013–2017	ZIP
# Federally Qualified Health Centers (# health center service delivery sites)	HRSA, 2019	Point
% Mammography (% female Medicare beneficiaries age 67–69 who had ≥1 mammogram over a 2-year period)	DAHC, 2015	HSA
% Preventable hospital stays (% hospitalizations of Medicare beneficiaries for ambulatory care-sensitive conditions)	DAHC, 2015	HSA
% Commute by bike (% commuting workers who use bicycles to commute)	ACS, 2013–2017	ZIP
% Commute by public transit (% commuting workers who use public transit to commute)	ACS, 2013–2017	ZIP

Acronyms: ACS (American Communities Survey), DAHC (Dartmouth Atlas of Health Care), GED (general educational development), HSA (hospital service area), HRSA (Health Resource and Services Administration).

**Table 2 ijerph-16-01786-t002:** Stanford well for life participant characteristics and overall well-being score (SWLS).

Covariate	Mean	SD	*n*
Age			
18–24	63.2	12.0	669
25–34	65.5	11.9	741
35–44	65.9	12.0	495
45–54	67.5	12.2	375
55–64	69.9	12.8	449
65+	72.5	10.6	397
(missing)	68.5	11.8	102
Gender			
Female	66.9	12.4	2291
Male	67.0	11.7	907
Other	59.3	16.2	30
Race			
White	67.4	12.7	1868
All Other Asian American	65.2	12.0	189
Black/African American	66.8	11.8	130
Chinese American	67.2	11.3	336
Native American/Native Alaskan	62.6	12.4	64
Other Race	66.6	12.2	172
Pacific Islander	65.6	10.6	61
(missing)	66.1	11.8	408
Marital status			
Single	63.8	12.3	1218
Married	69.9	11.7	1359
Partnered	65.8	11.9	285
Widowed divorced or separated	67.0	12.3	317
(missing)	64.7	12.4	49
Educational attainment			
High school	62.1	12.3	248
Some college	62.9	13.3	570
Bachelor’s degree	66.8	11.8	966
Graduate degree	70.0	11.4	1113
(missing)	67.3	11.8	331
Employment status			
Working	67.0	12.1	2079
Retired	72.4	10.9	315
Student	64.1	11.6	571
Other	65.0	14.8	254
(missing)	67.3	11.5	9
Household size			
Live alone	66.7	12.2	497
2-person	68.8	12.0	1048
3-person	65.0	12.6	636
4-person	66.5	12.1	607
5-person	66.4	12.3	196
>5-person	65.0	12.3	168
(missing)	65.6	12.7	76
Household income			
<$30k	61.7	13.2	286
$30k–$49k	63.9	12.0	275
$50k–$99k	65.0	12.1	807
$100k–$145k	67.4	12.1	556
$150k–$249k	69.3	11.5	636
>$250k	70.3	11.4	509
(missing)	68.3	13.3	159
Times moved in last 5 years			
None	68.9	12.3	1360
Once	65.9	11.9	743
More than once	65.1	12.2	1076
(missing)	64.9	12.8	49

**Table 3 ijerph-16-01786-t003:** Average SWLS Score for Stanford Well for Life Participants by Neighborhood Factor Quantile (lowest = Q1 to highest = Q5).

Neighborhood Factor	Mean SWLS					
	Q1	Q2	Q3	Q4	Q5	Missing
% Black/African American	66.4	68.5	66.3	67.1	66.1	-
% 9–12th grade education	67.3	68.5	67.2	65.2	66.2	-
% High school grad./GED	67.2	68.4	67.3	67.1	64.3	-
% Bachelor’s degree	64.5	67.3	67.3	67.7	67.2	-
% Divorced	66.7	68.0	68.1	66.5	65.0	-
Median household income	65.2	65.9	66.9	67.9	69.0	65.8
% Unemployed	68.0	67.2	67.1	67.3	64.7	-
% Child poverty	67.9	67.6	67.0	67.9	63.9	63.9
# Federally Qualified Health Centers *	67.4	65.6	65.7	-	-	-
% Mammography *	65.1	65.8	68.4	67.6	-	-
% Preventable hospital stays	67.7	68.6	67.7	65.0	65.8	-
% Commute by bicycle	65.3	67.5	66.9	67.7	66.9	55.9
% Commute by public transit	65.6	66.9	68.1	68.2	65.6	55.9

* Note: neighborhood factor was deliberately grouped into fewer than five quantiles.

**Table 4 ijerph-16-01786-t004:** Coefficient estimates by quantile group: Neighborhood-level factors and overall well-being.

Neighborhood Factor	Q1	Q2	Q3	Q4	Q5	*p*-Value *
Race: % Black/African American	ref	10.5	4.4	−7.4	10.6	0.050
Education: % 9–12 grade, no degree	ref	−9.7	−14.4	−18	−11.8	0.019
Education: % high school diploma/GED	ref	−3.2	4.6	12.2	15.6	0.138
Education: % bachelor’s degree	ref	13.9	13	17	19.1	0.045
Median income	ref	13.4	14.8	18.6	17.8	0.001
% Mammography	ref	−5.9	−5.6	−18.6		0.035
Preventable hospital stays per 1k	ref	−19.8	−20.7	−25.6	−22.1	0.001
% Commute by public transit	ref	−2	4.1	3.4	−11.7	0.030

* *p*-values determined by Type II sums of squares tests for ANOVA effects.

**Table 5 ijerph-16-01786-t005:** Significant differences in well-being outcomes between neighborhood factor quantile groups (determined by Type II sums of squares tests for ANOVA effects, reported as *p*-values).

Neighborhood Factor	SWLS	CR	EM	FI	PH	LI	PU	SP	SS	SC	ST	# Sig. Domains
Race: % Black/African American	0.050		0.006	0.059	0.002						0.022	3
Edu: % 9–12 grade, no degree	0.019		0.003		0.046	0.015			0.009		0.001	5
Edu: % high school diploma/GED	0.138				0.312		0.036					1
Edu: % bachelor’s degree	0.045	0.07	0.024	0.001	<0.001	0.017		<0.001			0.005	6
% Divorce					0.03						0.004	2
Median income	0.001		<0.001	0.014		<0.001			0.001	<0.001	0.123	5
% Children in poverty								0.001		0.057		1
# Federally Qualified Health Centers		0.113									0.063	0
% Mammography	0.035			0.004	0.057	0.062		0.024				2
Preventable hospital stays	0.001	0.017	0.002	<0.001	0.001				0.028	0.017	<0.001	7
% Commute by bike				0.047						0.004		2
% Commute by public transit	0.03		<0.001	0.013	0.286	0.119		0.127	0.102	<0.001	0.001	4
# Sig. neighborhood factors	6	1	6	6	5	3	1	3	3	4	6	
*r2*	0.064	0.008	0.042	0.075	0.081	0.132	0.029	0.039	0.041	0.039	0.058	
*ICC*	0.014	0.000	0.000	0.019	0.027	0.045	0.012	0.012	0.012	0.000	0.012	
*AIC*	44829	13146	18593	32015	19904	34280	14247	14605	23661	21447	16768	
Valid *n*	3175	3195	3170	3201	3182	3178	3187	3202	3196	3193	3159	
Moran I (model residual)	−0.001	−0.001	−0.001	−0.001	−0.001	−0.001	−0.001	0.000	0.000	−0.001	−0.001	
*p*-value	0.731	0.803	0.796	0.886	0.816	0.656	0.853	0.431	0.625	0.903	0.759	
Moran I (outcome)	0.005	0.001	0.005	0.007	0.003	0.018	0.001	0.006	0.003	0.006	0.005	
*p*-value	<0.001	0.02	<0.001	<0.001	<0.001	<0.001	0.001	<0.001	<0.001	<0.001	<0.001	

Abbreviations: CR (exploration and creativity), EM (emotional and mental health), FI (financial security and satisfaction), FQHC (Federally Qualified Health Center), GED (General Educational Development), LI (lifestyle and daily practices), PH (physical health), PU (purpose and meaning), SP (spirituality and religiosity), SS (sense of self), SC (social connectedness), ST (stress and resilience), SWLS (Stanford Well for Life Scale).

**Table 6 ijerph-16-01786-t006:** Significant differences in well-being outcomes between neighborhood factor quantile groups, adjusted for individual-level covariate (determined by Type II sums of squares tests for ANOVA effects, reported as *p*-values).

Variable	SWLS	CR	EM	FI	PH	LI	PU	SP	SS	SC	ST	# Sig. Domains
Neighborhood factors												
Race: % Black/African American	0.13		0.033	0.242	0.038							2
Edu: % 9–12 grade, no degree	0.019		0.006		0.046				0.006	0.02	0.019	5
Edu: % high school diploma/GED					0.019							1
Edu: % bachelor’s degree				0.004	<0.001			<0.001		0.036		4
% Divorce				0.025		0.136						1
Median income						0.004						1
% Children in poverty		0.012			0.192			0.064			0.106	1
# Federally Qualified Health Centers	0.132			0.094						0.142		0
% Mammography				0.002	0.194	0.054		0.007				2
Preventable hospital stays				0.004	0.124	0.032						2
% Commute by bike				<0.001		0.018					0.048	3
% Commute by public transit	0.08		0.02								0.072	1
Individual factors												
Age	<0.001	0.025	0.004	<0.001	<0.001			<0.001		<0.001		6
Educational attainment	<0.001	<0.001	0.05	<0.001	<0.001	<0.001	<0.001		<0.001	<0.001	<0.001	9
Gender	0.035		<0.001	0.047	<0.001	0.009		<0.001			<0.001	6
Household size	<0.001		<0.001	<0.001	0.05	0.035		<0.001	0.002	0.011	0.001	7
Household income	<0.001		<0.001	<0.001	<0.001	0.027		0.045	<0.001	<0.001	<0.001	8
Marital status	0.004		0.036	0.101	0.006	<0.001	<0.001	0.006		<0.001		6
Recent residential moves				0.148		0.001			0.005	0.125		2
Race/ethnicity				0.038	0.002	<0.001	0.041	<0.001		0.055	<0.001	6
Employment status			0.006	<0.001	0.033	0.025		0.002	0.013	0.046	<0.001	8
# Sig. neighborhood factors	1	1	3	5	4	3	0	2	1	2	2	
*r2*	0.167	0.029	0.132	0.264	0.178	0.226	0.066	0.111	0.086	0.164	0.140	
*ICC*	0.011	0.000	0.001	0.013	0.022	0.013	0.011	0.004	0.003	0.016	0.005	
*AIC*	33306	9851	13778	23214	14765	25290	10598	10858	17645	15809	12436	
Valid *n*	2382	2397	2378	2388	2386	2380	2388	2401	2397	2390	2373	
Moran I (model residual)	−0.001	<0.001	−0.001	−0.001	−0.001	−0.001	−0.001	<0.001	<0.001	−0.001	<0.001	
*p*-value	0.649	0.484	0.784	0.883	0.77	0.745	0.836	0.537	0.384	0.831	0.449	
Moran I (outcome)	0.006	<0.001	0.006	0.009	0.003	0.019	0.001	0.008	0.004	0.008	0.008	
*p*-value	<0.001	0.157	<0.001	<0.001	<0.001	<0.001	0.014	<0.001	<0.001	<0.001	<0.001	

Abbreviations: CR (exploration and creativity), EM (emotional and mental health), FI (financial security and satisfaction), FQHC (Federally Qualified Health Center), GED (General Educational Development), LI (lifestyle and daily practices), PH (physical health), PU (purpose and meaning), SP (spirituality and religiosity), SS (sense of self), SC (social connectedness), ST (stress and resilience), SWLS (Stanford Well for Life Scale).

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
