# Peer review of "Understanding Where We Are Well: Neighborhood-Level Social and Environmental Correlates of Well-Being in the Stanford Well for Life Study"

_ijerph, 2019, doi:10.3390/ijerph16101786_

Round 1
Reviewer 1 Report
The paper is well written and deals with a very interesting topic. However, I found some unclear parts in the manuscript. Therefore, the following comments can be considered in order to improve the manuscript.
Broad comments
Comment 1: Mixed models are used and the results are clearly presented. However, the model diagnostics have not been mentioned in the paper. How do the models perform? You assume that the response variables are continuous – also do you have outliers? Please expand the limitations related to this. In the limitations you can add that the nature of the variables can be taken into account in future work. I guess they are ordinal.
Comment 2: I have got another query related to the modelling strategies used in the paper. At some point, you include the random effects; how large (or small) are the groups in your sample? According to statistical modelling literature, there might be some issues in mixed models fitting when the sample size in the group is small. Furthermore, you could report the intra-class correlation.
Comments 3: Some more details on the design of the data used in the paper are necessary. Do they provide a sort of weights to take into account for the non-response?
Specific comments
Line 15: small n is used across all the paper to denote the sample size, please be consistent and use n for sample size.
Line 20: What do you mean by “a recent”?
Line 27: I am happy with grouping the observations into quantiles – but why are you doing that? Perhaps mention this point somewhere in the paper.
Line 37: please spell-out GED.
Line 66: I am a little bit uncomfortable with “selected segments of well-being measurement tools”. Is there any other way to rephrase this?
Line 128: On the assumption that the outcome variables are continuous please Comment 1 above.
Line 167: Please add a reference of Tukey transformations. Did you use Tukey's Ladder of Powers? In thiscase the reference can be the following Tukey, J. W. (1977). Exploratory Data Analysis. Addison-Wesley, Reading, MA. But please check with the software that might have the reference in it.
Line 175: Please also add and discuss this as limitation in the final discussion.
Line 176: Please give a reference for the inclusion of a spatial autocorrelation process. Also, you could justify this with some spatial descriptive statistics.
Line 178: Please do not use the word “completed” – you can write something like “The linear mixed models were estimated […]” or use the word “performed”.
Line 179: “Two models were constructed”.
Line 200: See also Comment 3 above. You may want to add more on the group sample sizes (min, max, etc). This is important when you include the random effects in the linear model.
Line 205: Table 2 N should be n if this denotes the sample size.
Line 222 and 223: You can expand this point a little bit more.
Line 303: Please be careful with that sentence about inference. You first say that you make statistical inference (“by testing for significant differences across[…]”) and they you say that you do not. Also the point of non-linear effects is very interesting.
Line 313: the self-selection bias is a very important limitation and this is good you mentioned it there. Maybe, you can add something about it in the introduction.
Author Response
REVIEWER #1
The paper is well written and deals with a very interesting topic. However, I found some unclear parts in the manuscript. Therefore, the following comments can be considered in order to improve the manuscript.
Thank you for your careful and constructive comments on this paper. We appreciate the feedback and hope that our comments and revisions below will address your concerns.
Broad comments
(1) Comment 1: Mixed models are used and the results are clearly presented. However, the model diagnostics have not been mentioned in the paper. How do the models perform?
a. Thank you for raising this important issue. We have now provided model diagnostic and performance criteria (Akaike Information Criterion and Intra-Class Correlations) in Tables 4 and 5. We describe these calculations in Section 3.5:
i. “To assess the amount of variance explained by the ZIP code groupings, intra-class correlation coefficients were calculated, and marginal and conditional r-squared values were calculated to assess variance with and without the ZIP-level random effect [29–31].”
(2) You assume that the response variables are continuous - Please expand the limitations related to this. In the limitations you can add that the nature of the variables can be taken into account in future work. I guess they are ordinal.
a. Thank you for raising this important issue. We have added several sentences to the Limitations section about data assumptions, including some language about our continuous treatment of variables:
i. “First, while SWLS domain scores are treated as continuous for the purpose of this exploratory study, some domains were generated from single questions, making them ordinal in nature, which may be more fully investigated in future studies.”
(3) also do you have outliers?
a. We have revised the analysis to address potentially influential outliers. Please see Section 3.5, as well as the supplemental materials, where the number of omitted outlier cases is described.
i. From Section 3.5: “Prior to this procedure, outliers (defined as observations having a standardized residual distance greater than 2.5 from 0) were identified based on the initial partial and full model fits and removed [26].”
(4) Comment 2: I have got another query related to the modelling strategies used in the paper. At some point, you include the random effects; how large (or small) are the groups in your sample? According to statistical modelling literature, there might be some issues in mixed models fitting when the sample size in the group is small.
a. The number of level 2 groups (i.e. ZIPs) and summary statistics for the number of level 1 observations (i.e., participants) within groups are reported in Section 3.1. Indeed, some concern has been raised in the multilevel modeling literature about group size, though these issues are heightened when group size is small; here, we have a relatively large group size (n=485), with an average number of participants per group of 7.1 (min=1, max=584). One attractive feature of multilevel models is their ability to incorporate information from relatively sparse groups into an overall estimate, while also drawing information from more information-rich groups.
b. For example, Gelman and Hill (2006, p.254) describe a scenario in which radon is sampled from homes (level 1) in counties (level 2) in Minnesota: “The weighted average (12.1) reflects the relative amount of information available about the individual county, on one hand, and the average of all the counties, on the other. Averages from counties with smaller sample sizes carry less information, and the weighting pulls the multilevel estimates closer to the overall state average. […] Averages from counties with larger sample sizes carry more information, and the corresponding multilevel estimates are close to the county averages. […] In intermediate cases, the multilevel estimate lies between the two extremes.”
(5) Furthermore, you could report the intra-class correlation.
a. Thank you for this suggestion. We now report intra-class correlation in the output tables.
(6) Comments 3: Some more details on the design of the data used in the paper are necessary. Do they provide a sort of weights to take into account for the non-response?
a. We have added additional details about the assumptions made about the SWLS dataset to the Limitations section, including language about the possible future use of weighting schemes such as the one offered by the Reviewer:
i. From the Limitations section: “Several data assumptions were made that are important limitations. First, while SWLS domain scores are treated as continuous for the purpose of this exploratory study, some domains were generated from single questions, making them ordinal in nature, which may be more fully investigated in future studies. Second, we assume data to be missing at random, which may be a source of bias should significant patterns of missingness exist. Future studies might incorporate a weighting schemes based on participant non-response, which is not currently a feature of the SWLS dataset.”
Specific comments
(7) Line 15: small n is used across all the paper to denote the sample size, please be consistent and use n for sample size.
a. Thank you for catching this error. We have revised the manuscript for consistency in naming/labeling.
(8) Line 20: What do you mean by “a recent”?
a. We have revised this sentence to clarify that the study was published in 2018.
(9) Line 27: I am happy with grouping the observations into quantiles – but why are you doing that? Perhaps mention this point somewhere in the paper.
a. We have added some additional language in Section 3.4 to clarify that our use of quantile groups is in line with the methods used by the comparison study by Roy and colleagues (2018). Please also see Lines XXX in the Discussion section, which further describe how quantile groupings were used to investigate variation across levels of the different neighborhood factors in an exploratory manner, rather than as continuous measure.
i. “Finally, by testing for significant differences across groupings of neighborhood factors (as in Roy et al. [7]), rather than seeking to draw more direct “dose-response” inferences, we allowed for the possibility that certain effects of neighborhood characteristics on well-being are non-linear (e.g., both extremely low and extremely high values exhibited a positive relationship with well-being, while mid-range values had a negative relationship).”
(10) Line 37: please spell-out GED.
a. Thank you for catching this omission. We have revised the manuscript accordingly.
(11) Line 66: I am a little bit uncomfortable with “selected segments of well-being measurement tools”. Is there any other way to rephrase this?
a. We have revised this sentence to read: “have primarily used selections of well-being survey instruments”
(12) Line 128: On the assumption that the outcome variables are continuous please Comment 1 above.
a. We have added language to the Limitations section describing our continuous treatment of ordinal variables for certain domain scores.
i. From the Limitations section: “First, while SWLS domain scores are treated as continuous for the purpose of this exploratory study, some domains were generated from single questions, making them ordinal in nature, which may be more fully investigated in future studies.”
(13) Line 167: Please add a reference of Tukey transformations. Did you use Tukey's Ladder of Powers? In this case the reference can be the following Tukey, J. W. (1977). Exploratory Data Analysis. Addison-Wesley, Reading, MA. But please check with the software that might have the reference in it.
a. Yes, thank you for providing this citation. We have clarified this in the text and added the citation to the References.
(14) Line 175: Please also add and discuss this as limitation in the final discussion.
a. We have added language about data assumptions, including assumptions about missingness, to the Limitations section.
i. “Several data assumptions were made that are important limitations. First, while SWLS domain scores are treated as continuous for the purpose of this exploratory study, some domains are generated from single questions, making them ordinal in nature. Second, we assume data to be missing at random, which may be a source of bias should significant patterns of missingness exist.”
(15) Line 176: Please give a reference for the inclusion of a spatial autocorrelation process. Also, you could justify this with some spatial descriptive statistics.
a. Language has been added to Section 3.5 to describe various tests now performed for spatial autocorrelation (Moran’s I), with the results displayed in Tables 5 and 6. Please see also our response to Reviewer 2 (item #1).
(16) Line 178: Please do not use the word “completed” – you can write something like “The linear mixed models were estimated […]” or use the word “performed”.
a. We have revised this sentence to use the word “performed,” as suggested.
(17) Line 179: “Two models were constructed”.
a. We have revised the sentence as suggested.
(18) Line 200: See also Comment 3 above. You may want to add more on the group sample sizes (min, max, etc). This is important when you include the random effects in the linear model.
(19) Line 205: Table 2 N should be n if this denotes the sample size.
a. Thank you for catching this error – we have revised for consistency throughout the manuscript.
(20) Line 222 and 223: You can expand this point a little bit more.
a. We have expanded this sentence to read: “While percent with partial high school completion is included in a model adjusted for individual-level covariates, the association with well-being is no longer significant, indicating that while it contributes to a better-fitting model (i.e., one with a lower AIC), it does not have a strong relationship with the main outcome variable”
(21) Line 303: Please be careful with that sentence about inference. You first say that you make statistical inference (“by testing for significant differences across[…]”) and they you say that you do not. Also the point of non-linear effects is very interesting.
a. We have revised this sentence to read: “Finally, by assessing differences across quantile groupings of neighborhood factors (as in Roy et al. [7]), rather than seeking to draw more precise “dose-response” inferences, we allowed for the possibility that certain effects of neighborhood characteristics on well-being are non-linear (e.g., both extremely low and extremely high values exhibited a positive relationship with well-being, while mid-range values had a negative relationship).”
(22) Line 313: the self-selection bias is a very important limitation and this is good you mentioned it there. Maybe, you can add something about it in the introduction.
a. Thank you for this suggestion. We have added an additional line to the Introduction: “Additional challenges arise when considering the potential sorting of individuals into neighborhoods that might suit their preexisting behaviors or traits [5].”
Reviewer 2 Report
In this manuscript the authors investigated the effects of neighborhood traits on individual well-being. As the authors said, they made contributions by using multi-dimensional measures for individual well-being and defining the neighborhood traits at the zip code level.
My major concerns are about the methods they have used. First, while the authors talked about spatial autocorrelation, they did not report the value of Moran’s I for average well-being of participants living in adjacent zip code areas, especially those in California. If the Moran’s I is large enough, for example, greater than 0.6, the authors should adopt spatial regression models. Second, the authors were analyzing a multiple level case. Participants within a zip code area tended to have similar well-being values than those from other zip code areas. Therefore, the authors should include the average well-being of all other participants in a zip code to control for the higher level effect.
There are a couple of minor issues.
(1) When “the Stanford WELL for Life Scale (SWLS)” first appeared in line 80 on page 2, there should be a reference for it.
(2) Contents in paragraph 5 on page 2 (lines 80-89) and paragraph 3 on page 3 (lines 114-129) were largely duplicated.
Author Response
REVIEWER #2
In this manuscript the authors investigated the effects of neighborhood traits on individual well-being. As the authors said, they made contributions by using multi-dimensional measures for individual well-being and defining the neighborhood traits at the zip code level.
Thank you for your careful and constructive comments on this paper. We appreciate the feedback and hope that our comments and revisions below will address your concerns.
(1) My major concerns are about the methods they have used. First, while the authors talked about spatial autocorrelation, they did not report the value of Moran’s I for average well-being of participants living in adjacent zip code areas, especially those in California. If the Moran’s I is large enough, for example, greater than 0.6, the authors should adopt spatial regression models.
a. Thank you for this suggestion. We have now incorporated several Moran’s I tests for spatial autocorrelation of the outcomes and model residuals. These tests are described in Section 3.5, and the results are conveyed in Tables 5 and 6.
i. From Section 3.5: “Spatial autocorrelation of the primary outcome variable was assessed at the level of participants (individual well-being scores) and using Morans’ I test statistic for spatial autocorrelation [20,21], which indicated significant geographic clustering of similar outcome values.”
b. While the model residuals did not appear to be spatially autocorrelated, significant p-values were obtained in Moran’s I tests for outcome variables (now reported in Tables 5 and 6). We accounted for this by introducing a spatial correlation structure as part of the multilevel model:
i. From Section 3.5: “To account for this spatial autocorrelation of the outcome variable, a Gaussian correlation structure was specified using latitude and longitude coordinates of ZIP code centroids. Spatial autocorrelation was then assessed with Moran’s I test for residuals from the full linear mixed model for the main well-being outcome. All linear mixed modeling was performed using the “lme” function from the “nlme” R package [22–25], and Moran’s I tests were performed using the “moran.test” function from the “spdep” R package [26]. These spatial statistics and their attendant p-values are reported in Tables 5 and6.”
(2) Second, the authors were analyzing a multiple level case. Participants within a zip code area tended to have similar well-being values than those from other zip code areas. Therefore, the authors should include the average well-being of all other participants in a zip code to control for the higher level effect.
a. As we described in our response to Reviewer #1 (Comment #4), a feature of the multilevel model is its ability to incorporate information from both data-rich groups and those that are more sparse. Applying an additional weighting scheme such as the one suggested would potentially introduce an additional source of bias, given the variable sizes of the groups in our study. Additionally, we account for within-group correlations of the outcome variable with the spatial correlation structure described in #1 above, and adopt a more conservative restricted maximum likelihood parameter estimation (“REML”) to partially account for the unbalanced nature of the higher level groups.
i. From Section 3.5: “With the covariates identified in stepwise model selection, partial and full models were refit using restricted maximum likelihood parameter estimation to partially account for the unbalanced nature of our higher-level groups (i.e. ZIPs).”
(3) When “the Stanford WELL for Life Scale (SWLS)” first appeared in line 80 on page 2, there should be a reference for it.
a. Thank you for this suggestion. We have moved the appropriate reference so that is more clear for the readers.
(4) Contents in paragraph 5 on page 2 (lines 80-89) and paragraph 3 on page 3 (lines 114-129) were largely duplicated.
a. Thank you for catching this duplication. We have revised the Introduction to more briefly mention SWLS and instead use the Materials and Methods section (3.1) to provide a more complete description.
Round 2
Reviewer 2 Report
The authors have done a good job responding to my prior concerns. I now find the manuscript to be much improved and the explanation much more reasonable. I think the paper does a strong job of extracting useful insight and it makes a definite contribution. Well done!